# Contemporary climatic analogs for 540 North American urban areas in the late 21st century

Matthew C. Fitzpatrick [1] & Robert R. Dunn[2,3]

A major challenge in articulating human dimensions of climate change lies in translating global climate forecasts into impact assessments that are intuitive to the public. Climate-analog mapping involves matching the expected future climate at a location (e.g., a person's city of residence) with current climate of another, potentially familiar, location - thereby providing a more relatable, place-based assessment of climate change. For 540 North American urban areas, we used climate-analog mapping to identify the location that has a contemporary climate most similar to each urban area's expected 2080's climate. We show that climate of most urban areas will shift considerably and become either more akin to contemporary climates hundreds of kilometers away and mainly to the south or will have no modern equivalent. Combined with an interactive web application, we provide an intuitive means of raising public awareness of the implications of climate change for 250 million urban residents.

[1] University of Maryland Centre for Environmental Science, Appalachian Laboratory, 301 Braddock Road, Frostburg, MD 21532, USA. [2] Department of Applied Ecology, Campus Box 7617, North Carolina State University Campus, Raleigh, NC 27695, USA. [3] Natural History Museum of Denmark, University of Copenhagen, Kobenhavn DK-2100, Denmark. Correspondence and requests for materials should be addressed to M.C.F. (email: mfitzpatrick@umces.edu)

Within the lifetime of children living today, the climate of many regions is projected to change from the familiar to conditions unlike those experienced in the same place by their parents, grandparents, or perhaps any generation in millenia[1,2]. While scientists share great concern for the expected severe impacts of climate change, the same is not necessarily true of the general public[3-5]. At the same time, decision makers have not formalized climate adaptation plans for a large proportion of major cities[6], and existing efforts often are considered insufficient to avoid social, environmental, and economic consequences of climate change[7].

Disconnects between the potential threats of climate change and societal action arise from multiple factors[4,5,8], but changing how people perceive and conceptualize climate change is considered key to improving public engagement[4,5,8]. For example, it is difficult for people to identify with the abstract, remote, descriptive predictions of future climate used by scientists (e.g., a 3 °C increase in mean global temperature). Translating and communicating these abstract predictions in terms of present-day, local, and concrete personal experiences may help overcome some barriers to public recognition of the risks (and opportunities) of climate change[9,10]. Given that most humans reside in urban areas and urban populations are considered highly sensitive to climate change[11], it is important to assess what climate change could mean for urban areas and to communicate the magnitude and uncertainty of these expected changes in intuitive ways.

Climate-analog mapping is a statistical technique that quantifies the similarity of a location's climate relative to the climate of another place and/or time[12-15]. When considered in the context of assessing and communicating exposure to future climate change, climate-analog mapping can be viewed as a form of forecasting by analogy[16,17] that translates the descriptive and abstract (i.e., scientific forecasts of future climate) into the familiar (i.e., present-day climate of a known location). Veloz et al.[18] used climate-analog mapping to find contemporary climatic analogs for projected future climates for the U.S. state of Wisconsin, while Rohat et al.[19] used similar methods to quantify and communicate the implications of climate change for 90 European cities. Climate-analog mapping is gaining popularity as a means to communicate climate change impacts[20,21], and more robust methods for measuring climatic similarity between places and times have been recently developed[22].

Here we use climate-analog mapping and an interactive web application (available at https://tinyurl.com/urbanclimate) to characterize and communicate how climate change may impact the lives of a large portion of the populations of the United States and Canada. Collectively, the 540 urban areas we analyze in this study include approximately 250 million inhabitants, including >75% of the population of the United States and >50% of the population of Canada. For each urban area, we mapped the similarity between that city's future climate expected by the 2080s (mean of the period 2070–2099)[23] and contemporary climate (representative of mean conditions for 1960–1990)[24] in the western hemisphere north of the equator (Supplementary Figure 1). We identified climatic analogs using sigma dissimilarity[22], a statistical measure that accounts for correlations between climate variables, incorporates historical interannual climatic variability (ICV), and converts multidimensional climatic distances to percentiles of a probability distribution of these distances. A sigma dissimilarity equal to 0 (i.e., $0\sigma$) would indicate identical climates, or a perfect analog. We considered values of $\leq 2\sigma$ between an urban area's future climate and its most similar contemporary climate to be a representative analog. Values $>4\sigma$ represent extreme differences between future climate and contemporary climate within the study domain, which we interpret

as novel future climatic conditions[22] and a poor analog. In this sense, sigma dissimilarity serves as both an indicator of climate novelty and a measure of the strength of analogy between an urban area's future climate and its best contemporary climate match.

We calculated sigma dissimilarity using minimum and maximum temperature and total precipitation for the four climatological seasons (12 climate variables total). For 2080's climate, we selected two emission trajectories or Representative Concentration Pathways (RCPs)[25], unmitigated emissions (RCP8.5) and a mitigation scenario (RCP4.5)[26], and 27 different earth system models (ESMs), for a total of 2 RCPs × 27 ESMs = 54 future climate scenarios (Supplementary Table 1). Here we emphasize results for the ensemble means of 2080's climate calculated by averaging across the 27 climate projections for each RCP.

For each future climate scenario, we calculated sigma dissimilarity between each urban area's future climate and every contemporary climate pixel in the study domain. We mapped the resulting sigma values to create a climate similarity surface and identified the pixel with the minimum sigma dissimilarity. This pixel represents the best contemporary climatic analog to 2080's climate for that urban area and climate scenario, again noting that values $>2\sigma$ increasingly characterize novel climates rather than representative analogs.

We find that if emissions continue to rise throughout the 21st century, climate of North American urban areas will become, on average, most like the contemporary climate of locations 850 km away and mainly to the south, with the distance, direction, and degree of similarity to the best analog varying by region and assumptions regarding future climate. For many urban areas, we found substantial differences between future climate and the best contemporary climatic analog, underscoring that by the 2080s many cities could experience novel climates with no modern equivalent in the study domain. In addition to the summaries we report here, we visualize climate analogs for all 540 urban areas and 54 future climate scenarios using an interactive web-based application (available at https://tinyurl.com/urbanclimate) that provides a means to communicate abstract forecasts of future climate in terms that are more locally relevant to the nearly 250 million people who call these urban areas home.

## Results

**Contemporary climate analog example using Washington D.C.** We can use climate-analog mapping to ask: what location has a contemporary climate that is most similar to Washington D.C.'s expected climate in the 2080s? The climate similarity surfaces (i.e., maps of sigma dissimilarity) show that the contemporary climates most similar to 2080's climate in Washington D.C. reside in low elevations across the southeastern United States (Fig. 1). However, few pixels represent good climatic analogs (i.e., $<2\sigma$, see contour lines Fig. 1a). And these matches are present only for the mitigated emissions scenario (RCP4.5) that assumes that policies are put in place to limit emissions[26]. For RCP4.5, the pixel with the lowest sigma dissimilarity ($0.57\sigma$) is located near Paragould, Arkansas. For the unmitigated emissions scenario (RCP8.5), the scenario most in line with what might be expected given current policies and the speed of global action[27], the climate similarity surface shifts further south and climate novelty increases. Under this scenario, the pixel with the lowest dissimilarity ($2.89\sigma$) is located near Greenwood, Mississippi (Fig. 1b), but all locations exceed the $2\sigma$ threshold, which is to say none are a very good match.

**Contemporary climatic analogs for North American urban areas**. By the 2080s, and even given the optimistic mitigated emissions scenario (RCP4.5)[26], climate of North American urban

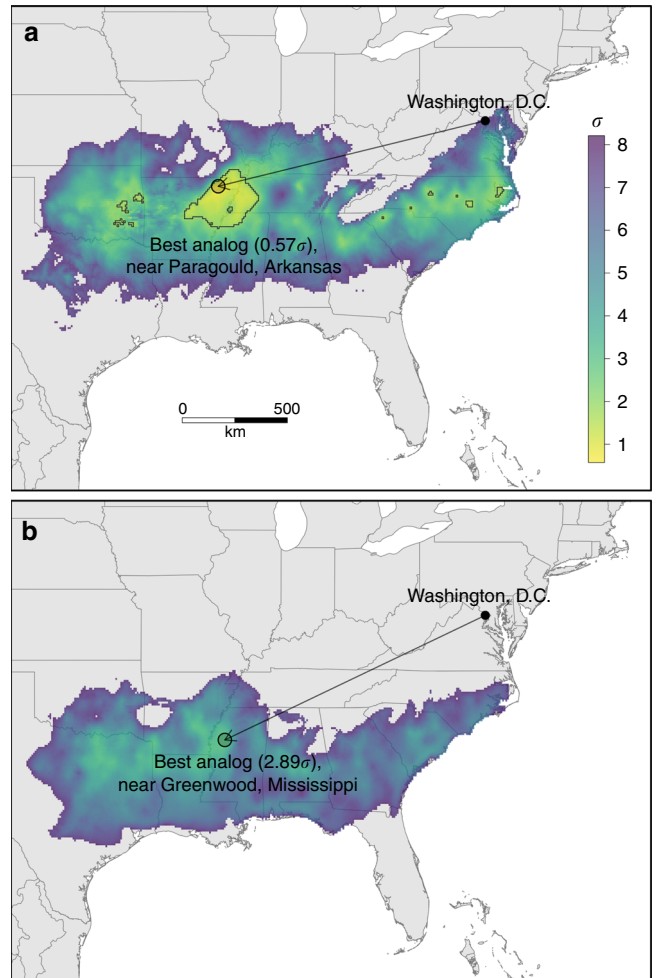

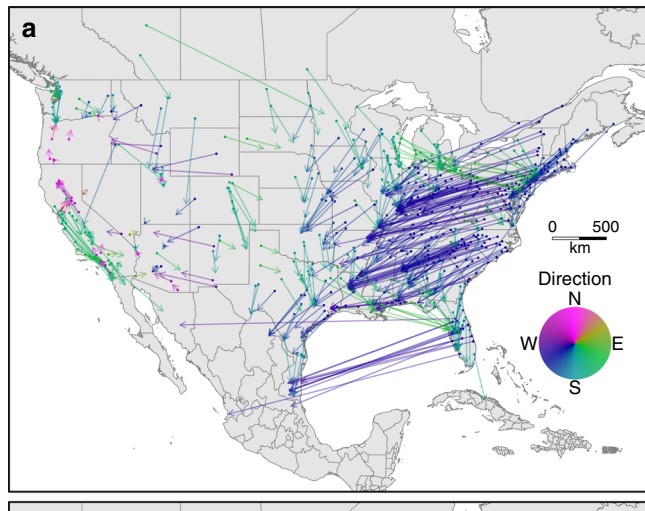

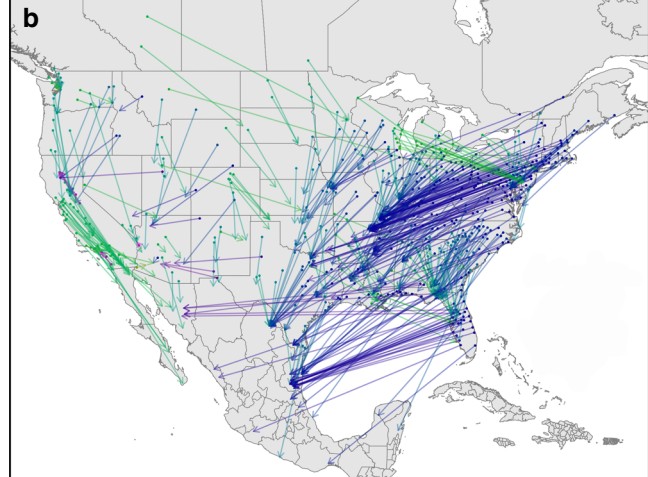

**Fig. 1** Climate analog maps for 2080's Washington DC. Shading indicates sigma dissimilarity for the ensemble mean of the 27 climate projections for **a** RCP4.5 and **b** RCP8.5. The arrow and circle highlight the location of the best contemporary climatic analog where sigma dissimilarity is minimized. Outlined pixels indicate locations with sigma dissimilarity <2 (present for RCP4.5 only)

**Fig. 2** Distance and direction to the best climatic analog. Arrows point from each urban area (filled circles) to the location of the best contemporary climatic analog for that urban area's climate in the 2080s based on the ensemble mean of 27 projections for **a** RCP4.5 and **b** RCP8.5. Shading indicates the initial bearing from each urban area to its best contemporary climatic analog

areas will feel substantially different than they do today, and in many cases unlike contemporary climates found anywhere in the western hemisphere north of the equator. In the eastern U.S., nearly all urban areas, including Boston, New York, and Philadelphia, will become most similar to contemporary climates located hundreds of kilometers to the south and southwest. Climates of most urban areas in the central and western U.S. will become most similar to contemporary climates found to the south or southeast (Fig. 2). Put another way, by the 2080s climate of cities in the northeast will tend to feel more like the humid subtropical climates typical of parts of the Midwest or southeastern U.S. today (warmer and wetter in all seasons, Supplementary Figure 2), whereas the climates of western cities are expected to become more like those of the desert Southwest or southern California (warmer in all seasons, with changes in the amount and seasonal distribution of precipitation, Supplementary Figure 3).

On average, the geographic distance from each urban area to its best contemporary climatic analog was nearly twice as large for RCP8.5 (849.8 km) as compared to RCP4.5 (514.4 km). In other words, the average urban dweller in the United States would have to drive nearly 1000 km to get to a climate like that likely to be experienced (under RCP8.5) in their city. The greatest geographic

distances between future climates of urban areas and their best contemporary climatic analogs were in the eastern U.S. This pattern is especially apparent for cities in Florida, for which best analogs were concentrated along the Gulf coast of Mexico (Fig. 2). The greater distances to the best analog for eastern urban areas likely reflect the influence topographic position on climate[22]. In short, in regions of high relief, such as portions of western North America, adjacent lower elevations can provide analogs to higher elevation climates that are expected to become warmer and drier. The average direction to the best analog was south-southwest and did not differ appreciably between the RCP4.5 and RCP8.5 emission scenarios (200.4° vs. 201.7°, respectively). However, for some west coast cities under RCP4.5 the closest analog was to the north (Fig. 2a), also likely reflecting the influence of topography on the location of the best climatic analog.

**Strength of analogy and climatic novelty**. The geographic location with the minimum sigma dissimilarity identifies the best contemporary climatic analog for a given city's future climate. However, the best contemporary climatic analog does not necessarily imply an analogous climate. For example, if the future climate of a given urban area is found to be novel (~≥4σ), then by

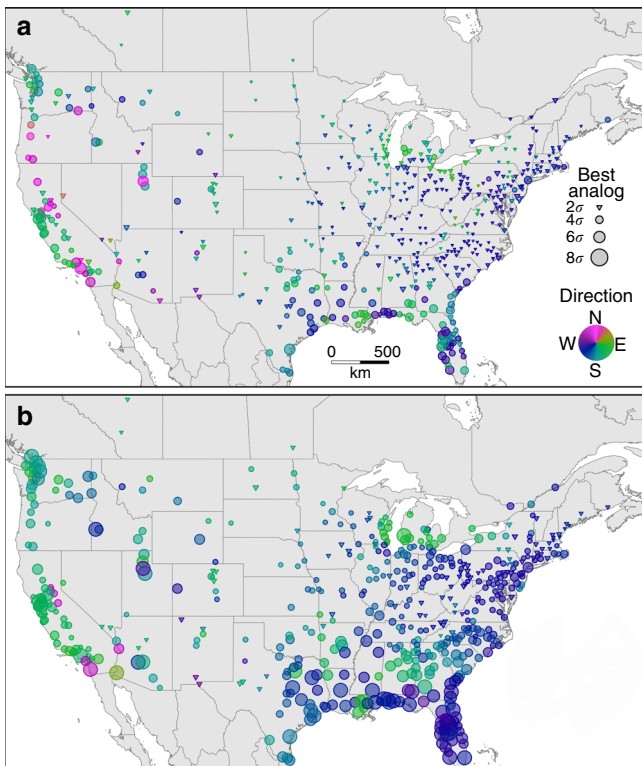

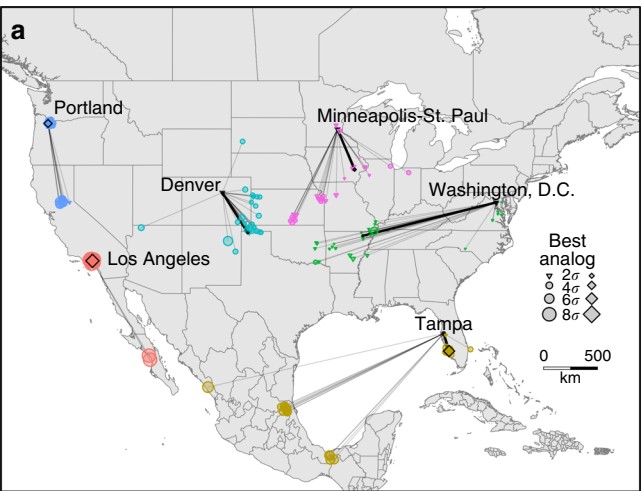

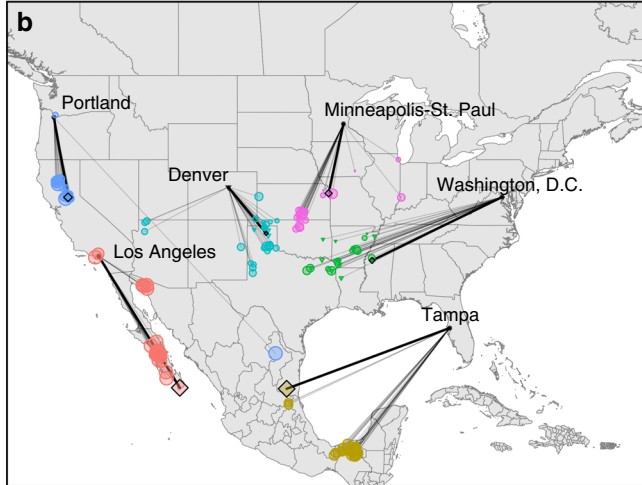

**Fig. 3** Strength of analogy for contemporary climatic analogs. Triangles indicate urban areas with representative contemporary analogs (<2σ); circles indicate increasingly poor analogs, with size of symbols scaled to sigma dissimilarity for the ensemble mean of the 27 projections under **a** RCP4.5 and **b** RCP8.5. Shading indicates the initial bearing from each urban area to its best contemporary climatic analog

**Fig. 4** Variation in climatic analogs by future climate scenario. For each of the six example cities, colored triangles and circles indicate the location of the best contemporary climatic analog to 2080's climate for the 27 future climate scenarios for **a** RCP4.5 and **b** RCP8.5. Triangles indicate representative contemporary analogs (<2σ) and circle size indicates increasingly poor analogs. Colored diamonds and bold lines indicate contemporary climatic analogs for the ensemble mean across the 27 individual projections

definition no location within the study domain possesses representative climatic conditions. For RCP4.5, we identified representative analogs (i.e., ≤2σ) for most (69.6%) urban areas west of the Rocky Mountains (Fig. 3a, triangles). For these urban areas and our threshold of 2σ, the best contemporary climatic analog can serve as a meaningful analogy for future climate. In contrast, for most urban areas along the western and southeastern coasts, there are no representative contemporary climatic analogs anywhere in the study domain, which likely reflects a combination of the lower topographic position of urban areas in these regions[22], the nature of forecasted climate change, and the pool of contemporary climates available within the study domain (Supplementary Figure 1) to serve as analogs. For cities with analogs >2σ, the most informative finding is not necessarily the contemporary climate of the best analog but rather the extreme dissimilarity and therefore novelty of the urban area's future climate. Future climate novelty becomes especially apparent for RCP8.5, for which there are extreme differences (>4σ) between expected future climate and contemporary climate for 42.7% of urban areas (Fig. 3b), with only 17% having an analog dissimilarity <2σ. Notably, current emissions are exceeding the RCP8.5 trajectory[27], and should these trends continue, the climate changes expected by the 2080s for RCP8.5 may arise earlier in the 21st century. If we continue on our current trajectory, the climate of many urban areas could become unlike anything present within the study domain, whereas keeping warming within the 1.5 °C goal set by the Paris Agreement could reduce the exposure of urban areas to climate novelty. Increasing the geographic extent of the study domain could identify better contemporary climatic analogs, though as the reference domain expands to include increasing

unfamiliar territory, the utility of forecasting by analogy decreases.

**Variation in climatic analogs across climate scenarios.** Prominent geographic patterns in the direction, distance, and degree of similarity to climatic analogs are evident for the ensemble means of each emission trajectory. However, variation becomes apparent when mapping the results for the 27 individual climate projections for different cities, highlighting variability in the location of best analogs for different realizations of future climate (Fig. 4; results for all urban areas available at https://tinyurl.com/urbanclimate). For example, the location and σ of the best contemporary analog to 2080's climate in Washington D.C. varies from northeastern Texas to northeastern Virginia depending on the future climate projection (Fig. 4). Like the climate-analog mapping approach itself, visualizing geographic variability in the location of the best analog provides an intuitive means of depicting underlying variability and uncertainty, as well as consensus, in climate forecasts. To the extent that climate analogs can inform planning and adaptation, cities for which there is high

geographic variability between projections may require policies better able to deal with high uncertainty and which emphasize resilience under climate change rather than adaptation to a particular climatic end state[28].

## Discussion

Given their large and growing populations, reliance on interconnected and in some cases aging infrastructure, and barriers to implementing coordinated climate adaptation efforts, urban areas are considered highly vulnerable to climate change[11]. We quantify and communicate the potential exposure of North American urban populations to changes in climatic conditions by identifying where on the landscape similar contemporary climates reside, but we ignore additional stressors, notably supplemental warming associated with urban heat island effects[29], sea-level rise, and extreme events. Nonetheless, by translating abstract statistical forecasts of future climate into something more akin to personal experience with present-day climates, climate-analog mapping can help communicate what future climate may feel like to urban residents in a broad sense, as well as the potential magnitude and nature of local climate change.

Meaningful visualization and communication are considered key components in raising public awareness of climate change[10,30], and approaches similar to those employed in this study have been found to be effective for conveying climate change information[31]. Successful policy implementation on climate change requires public support, which is predicated on fluent communication between scientists, the public, and decision makers. Therefore, in supplement to the summaries presented here, we developed an online, interactive web application (https://tinyurl.com/urbanclimate) that allows the general public, educators, decision makers, and stakeholders to explore in greater detail results for all 540 urban areas and 54 climate scenarios. The web application maps the locations of the best contemporary climatic analogs and their sigma dissimilarity, as well as climate similarity surfaces (as in Fig. 1) showing spatial variation in the strength of analogy between future and contemporary climates for each urban area. More research is needed to formally assess the efficacy of these visualizations for communicating climate change information and changing perceptions across a variety of potential end users.

Studies on the utility of analogs for informing policy[21] and assessing economic impacts[13] have been mixed. In addition to climate similarity, economic differences and social and political norms related to pro-environmental behavior and decision making are likely to influence the extent to which climatic analogs can advance policy, adaptation, and planning. Climate novelty further challenges the applied utility of climate analogs. We found that many urban areas do not have a representative analog within the study domain, especially if emissions continue unabated throughout the 21st century. Urban areas found to have novel climates generally corresponded to the same regions of high climate novelty as found by Mahony et al.[22] (i.e., the southeastern and western coast of the U.S.) using different datasets, which likely reflects the lower topographic position of urban areas in these regions. Climate novelty reduces our ability to use the past and present to inform the future, and in regions where climate novelty is expected to be high, the utility of forecasting by analogy is diminished as there are no representative analogs within the study domain for anticipating impacts. For these urban areas, existing infrastructure may need to meet the demands of climates that not only are changing but, in some cases, will no longer exist.

Beyond assessing exposure and communicating aspects of climate change itself, climate-analog mapping also can inform how climate change could impact agricultural and natural systems upon which humans depend. For example, climatic similarity between regions is often predictive of the success of introduced organisms[32]. In this sense, climate-analog mapping based on an ecologically relevant set of variables could be used to identify potential source regions for new, potentially problematic organisms, including weeds, insect pests, and diseases that could impact human health, wildlife, and agriculture. For example, our results suggest northeastern cities by the 2080s could support species now limited to the southeastern U.S. In instances where these new residents are disease vectors (e.g., Asian Tiger mosquitoes and West Nile Virus), urban areas could experience increased incidence of disease. However, as for other applications of climate analogs, climate novelty challenges our ability to use current patterns to anticipate potential ecological responses and threats[1].

Ultimately, what climatic analog analyses offer is not so much new models of the future but rather a means to communicate existing models such that their predictions are less abstract and psychologically distant and more local, experiential, and personal. It is difficult for individuals to detect and conceptualize gradual changes in climate, particularly where natural variability is high and when expected changes in climate are couched solely in numbers (mean temperatures, precipitation variability, and so forth). A crucial next step is to join with educators, psychologists, and social scientists to assess the extent to which climate-analog mapping can help increase climate change engagement and awareness[20,31].

## Methods

**Urban areas**. We quantified contemporary climatic analogs for 540 urban areas, including 530 in the United States and 10 in Canada. Urban area boundaries in the U.S. were based on the U.S. Census Bureau's 2010 TIGER/Line shapefiles[33]. Boundaries for Canadian urban areas came from the World Urban Areas dataset[34]. To ensure a large enough sample size for statistical analyses (where $n$ is the number of ~$1 \times 1$ km² pixels within the urban area), we selected urban areas with a geographic area ~>50 km². Urban areas were removed from further consideration if they lacked sufficient historical weather records or spatial heterogeneity in climate as necessary for calculating sigma dissimilarity. We also grouped some adjacent urban areas, for example, to combine suburbs with an adjacent city (e.g., Anchorage, Alaska and Northeast Anchorage, Alaska), resulting in 540 urban areas total.

**Identifying climatic analogs**. We used the sigma dissimilarity approach of Mahony et al.[22] to quantify the similarity of future climates (strength of analogy) of urban areas to contemporary climate. Sigma dissimilarity is derived from the Mahalanobis distance[35], which is a multivariate distance that is independent of the scales of the climate variables and which weights the contribution of individual climate variables to the statistical measure of distance according to their collinearity with all other variables (i.e., Mahalanobis distance corrects for the inflation of distance that occurs when variables are not orthogonal). Mahalanobis distances can be represented as probabilities of a chi distribution with $n$ degrees of freedom, where $n$ is the number of dimensions (climate variables) in which Mahalanobis distances are measured. Sigma dissimilarity is a multivariate $z$-score metric that represents the percentile of a given Mahalanobis distance within this chi distribution[22]. We calculated sigma dissimilarity using 12 seasonal climate variables: minimum and maximum temperature and total precipitation for the four climatological seasons (winter (DJF), spring (MAM), summer (JJA), and autumn (SON)). A sigma dissimilarity of two ($2\sigma$) represents the 95th percentile of the associated chi distribution, while $4\sigma$ represents the 99.994th percentile. Following Mahony et al.[22], we consider $2\sigma$ to represent a moderate degree of analog dissimilarity and therefore the upper threshold of what we viewed as a representative analog.

**Climate datasets**. Our calculation of sigma dissimilarity required three climate datasets, including contemporary climatic normals to represent the analog pool across the study domain, projections of future climate for each urban area, and estimates of historical ICV for each urban area. For contemporary climatic normals, we used monthly precipitation and temperature variables representative of the period 1960–1990 at 5 arc-minute resolution from the WorldClim dataset[24]. Choice of the reference time period used to represent contemporary climate is likely to influence climate analog analyses. Our use of the 1960–1990 as the baseline will not factor in recent changes in climate and therefore our estimates of climatic analogs would tend to be skewed accordingly. We cropped the contemporary climate data to the study domain bounded between equator to 80° N latitude and between 30° and 170° W longitude (Supplementary Figure 1).

Downscaled and bias-corrected projections[22] of future climate for the 2080s (30-year running mean of the period 2070–2099) at 30 arc-second resolution were obtained from the Consultative Group for International Agricultural Research Research Program on Climate Change, Agriculture and Food Security (http://www.ccafs-climate.org/)[36]. These future climate surfaces are downscaled and debiased using a delta method or change-factor method, whereby climate anomalies are calculated between present-day simulations and future simulations and the resulting anomalies are resampled to the desired resolution of the observational dataset using spatial interpolation techniques. These downscaled anomalies are then added to baseline current climate observations. See Ramirez-Villegas and Jarvis[23] for details. This method has been widely employed in climate change research given that it is simple and fast and therefore the most feasible for downscaling a large number of climate simulations. To describe a range of potential future climate conditions for the 2080s, we used 27 different ESMs (Supplementary Table 1) forced by two RCPs: RCP4.5 (a mitigated emissions scenario) and RCP8.5 (an unmitigated emissions scenario), developed as part of the Intergovernmental Panel on Climate Change Fifth Assessment Report[25]. In total, our analyses considered 54 future climate scenarios (27 ESMs × 2 RCPs). We selected a coarser resolution for the contemporary climate datasets (5 arc-minute vs. 30 arc-second for the future projections) to ease computational burdens.

We estimated historical ICV in the 12 temperature and precipitation variables for each urban area using NOAA weather station records. For each urban area, we used the rnoaa package[37] in R[38] and custom R scripts to find all weather stations within the urban area boundary that contained a complete time series of climatological records (monthly means of minimum and maximum temperature and total precipitation) for the contemporary reference period (1960–1990). If fewer than five weather stations within the urban area boundary met these requirements, stations outside the urban area boundary were searched, up to a maximum of 50 km from the urban area centroid, until five stations meeting the data requirements were found. Climate records were then averaged across stations using inverse distance weighting to down weight the contribution of records more distant from the urban area centroid. The monthly records were then aggregated to the four climatological seasons to match the contemporary climate normals and future climate projections.

**Statistical analyses**. For each urban area, we extracted climatic conditions projected for the 2080s from each of the 54 climate scenarios. We also averaged temperature and precipitation forecasts from each of the 27 scenarios across each RCP to create an ensemble mean projection for that RCP (as opposed to first calculating sigma dissimilarity for each future climate scenario and averaging the sigma dissimilarity values). Results based on these ensemble mean forecasts are emphasized here, whereas results for individual scenarios are available online at https://tinyurl.com/urbanclimate. We used the method and R scripts described in Mahony et al.[22] to convert the contemporary climate surfaces into sigma dissimilarity for each urban area and future scenario, which quantified the dissimilarity between the urban area's future climate and contemporary climate for each grid cell in the study domain, with appropriate scaling by historical ICV. We defined the best contemporary climatic analog to an urban area's future climate as the 5 arc-minute grid cell with the smallest sigma dissimilarity and we calculated the geographic distance and bearing from the urban area centroid to the centroid of this best-analog grid cell.

**Code availability**. Custom R code may be provided by M.C.F. upon request. For contact details, please consult the author's details stated above.

## Data availability

The datasets analyzed in the current study are publicly available from the sources referenced in the paper. Input or intermediate data may be provided by M.C.F. upon request.

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

## Acknowledgements
We thank G.F. and A.G. for help with visuals, and M.L. for help with Shiny app development. M.C.F. acknowledges funding from UMCES and NSF DEB-1257164. R.R.D. acknowledges support from the USGS Cooperative Agreement nos. G11AC20471, G13AC00405, and G15AP00153.

## Author contributions
M.C.F. designed the study, developed the data processing scripts, performed the analyses, developed and implemented the web application, and led the manuscript writing. R.R.D. co-designed the study and contributed to writing and editing the manuscript.

## Additional information

**Competing interests:** The authors declare no competing interests.

