## [Peer Review File · Nature Communications]

Reviewers' comments:

Reviewer #1 (Remarks to the Author):

This is a very welcome and interesting contribution that employs a state-of-the-art climate matching approach to explore future climate analogs of 540 urban areas in North America, under two emissions scenarios and 27 climate models. The paper is innovative and clearly push forward the field of climate analogs research. Such research is also particularly attractive and relevant for a lay audience, and has a very useful online tool that can be employed by stakeholders and the general public to explore future climate analogs of all the 540 cities. In short, I think this paper fits very well the high-quality standards and the scope of Nature Communications. Yet, I do have a number of issues with the current version, that I think need to be addressed.

Most of them relate to the fact that you oversell the potential applications of your approach as well as its results. For instance, you state in the abstract (L35) that “our results suggest that through the interactions with infrastructure, resources, and human health, climate change may have profound implications for urban populations”. This, in my opinion, does not hold true. This research identifies the climate analogs of 540 urban areas, based on temperature and precipitation only. Infrastructure, resources, health, etc. are not touched upon in the paper, hence such concluding (and very broad) statement should not be made. I suggest to stick to the main goal of this paper, which is (as you stated L33): “providing an intuitive means of raising public awareness and informing adaptation strategies”.

Hereunder are specific comments and questions that I would like you to address. These are mostly minor comments.

L19: why “vulnerability assessments”? What is the link with climate analogs? Are climate analogs really useful for vulnerability assessments?

L20: climate analogs have also been used to match the current climate of a location with the past climate of another location, e.g. see Beniston, 2013 (<https://doi.org/10.1002/joc.3804>). This might be worth mentioning.

L26: is RCP8.5 a business-as-usual scenario? It is often rather described as a high-end scenario. Also, for the readers that are not familiar with the RCPs, it might be good to avoid referring to the RCP in the abstract (such a paper might be of interest to a lay audience).

L30: in the same sentence you say “under business-as-usual emissions” and “varying by future climate scenario”, that is confusing. Please clarify.

L80: These are novel future climatic conditions for the region investigated only (North-America), but these may well exist elsewhere.

L88: This is a very well designed and useful online tool. In my opinion, it should be better advertised than through a simple link. I wonder to which extent this online tool can be embedded within the final manuscript online.

L120: Please explain shortly why the greatest geographic distances are expected for Eastern US cities. Is it because their future climate show larger changes, or because of their geographical location, or...?

L122: Fig2: is there a specific reason explaining climate analogs of some North-Western US cities are located on the North, unlike all the other cities' analogs? Would be nice to have this explained in the paper.

L135: Again, it would prove useful to have some more detailed explanations here. Why such geographical patterns? Why cities west of the Rocky Mountains have "better" climate analogs than cities of Western US and Florida? Is it due to different warming rates, or simply to the geographical border of the computational grid?

L144: Although the coordinates of the study domain are given in the Methods section, I find it difficult to picture it. It might be useful to have a map showing the study domain. If Fig.2 shows the study domain, then it could be stated in the legend.

L181: You quantified and communicated the potential magnitude of changes in climatic conditions, but not the magnitude of "future vulnerabilities" of the cities. These are two different things.

L186: In the same vein, one can doubt that such an approach facilitates the understanding of the nature of future climate impacts. Can you provide a bit more explanation here? An interesting reference to add is Jylhä et al. 2010 (<https://doi.org/10.1175/2010WCAS1010.1>), in which they actually tested (with a questionnaire) whether climate analogs are a useful tool for communicating future climate impacts.

L190: This discussion on the usefulness (and the limitations) of the climate analogs approach for adaptation/urban planning is missing two important papers: Kellett et al. 2015 (<https://doi.org/10.1016/j.landusepol.2014.11.022>), which points out the limitation of such an approach; and Hallegatte et al. 2007 (<https://doi.org/10.1007/s10584-006-9161-z>), in which authors employ the climate analog method to assess future economic impacts in urban areas.

L197: studi → study

L198-203: I think that this statement is a bit overselling the paper (and there is no need, as it is already very innovative). The climatic variables that you employed are not determinants for weeds, insect pests, diseases, etc, which also have a range of climatic conditions to which they can spread (and this is not reflected in the climate analog approach). I agree that applying the climate analogs for these issues is worth exploring (it has already been applied for forestry or wine for instance), but your paper does not do this. I would rather rephrase to reflect the potential applications of the climate analog approach, but with different climatic variables (i.e. oriented towards the goal of the

study, e.g. specific to a given type of agriculture, to a given disease, etc.). In the same vein, I would suggest to focus the discussion on the benefits of your approach for the design of urban adaptation strategies, climate-resilient infrastructure, etc (as touched upon in L188-197).

L233: Needs more clarification on this rather non-transparent weighting of the contribution of the 12 individual climate variables. Is there one variable that weighs much more than the other? How much dissimilarity for a given variable is allowed? For instance, if the min/max seasonal temperatures fit perfectly with the analog's temperature regime but the precipitation regimes show a very poor match, is the sigma dissimilarity still low? This needs some transparent clarification.

L257: are these projections bias-corrected? If not, this should be at least acknowledged (as well as the uncertainties associated with non-bias corrected climatic projections).

L280: what does "2080s" stand for? Is it 2070-2090? 2060-2100? Please clarify

L281: this is a critical point of the method. I initially thought that you computed the location of the climate analog for each climate model, and then employed a mapping approach to define the location that fits best the 27 climate analog of a given city (e.g. the location that minimizes the distance between each of the 27 analogs' location). It seems however that you averaged the climate data over the 27 ESM, and then defined afterwards the climate analog. This is still a valid approach, but you need to be more critical and to acknowledge that there is other ways of computing analogs for an ensemble of climate models.

Reviewer #2 (Remarks to the Author):

This article demonstrated present-day climatic analogs for 540 North American urban areas in 2080. Climate analogue has been used for more than a century and applied to future cities under a changing climate for several decades. The number of the target cities in this article are larger than previous articles but no scientific novelty is seen except for this large number. The target of the audiences is vague since there are many decision makers. Therefore, this article in the present form is not recommended for publication in this journal. There are also some scientifically unclear points mentioned below. By responding these points, this article will improve in scientific contents.

I do not understand why the distance and the cardinal direction is important in this context: "RCP8.5 (849.8 km) as compared to RCP4.5 (514.4 km)" and "the cardinal direction did not differ appreciably between the RCP4.5 and RCP8.5 emission scenarios (200.4° vs. 201.7° respectively)"

Methods

Both the Worldclim and NOAA station data are used in this study. The climatological mean values of the NOAA station data are not always the same. You should show the difference between the two climatologies.

How do you prepare the future climate values? Please describe it since the model has biases in all variables.

In search of present-day climate analogue, three variables: two air temperatures and precipitation. The climate analogue may be weakly influenced by precipitation. The two air temperatures are projected to rise by almost same degrees and do not provide independent information. It is better to use mean air temperature or to use weighting during the search of climate analogues.

These three combinations may be justified for a specific purpose such as crop productions, please provide the justification for the selection.

Why is the year of 2080 is selected? The climate analogue analysis should use the same temporal average period. The method in this study use present-day climatological values and single-year values but with mutli-model ensemble. The variability of multi-model ensemble is not ICV but the variability of multi-model ensemble or uncertainties in future climate projections. Please provide the justification if both are interchangeable.

In Fig. 4, you identify the climate analog for each model's projection. For this case, you compared observed climatology with projected single year value. Please provide the justification.

I believed that use of climatological values of each model provide more information about uncertainties of climate analogues in future climate projections.

Figures S1 and S2 shows large discrepancies in values in some cities between present-day climatological values and future projected values, especially in precipitation. These discrepancies are acceptable?

Reviewer #3 (Remarks to the Author):

Article summary and contribution:

The study identifies and maps climate-analogs for 540 urban areas in the United States and Canada. The authors find that under business-as-usual emissions scenarios, the climate of North American urban areas will shift, on average, 928km away and mainly to the south. The authors also link to a novel online interactive mapping tool to visualize the climate-analogs for each city.

As far as I know, this is the first large-n analysis of climate-analogs in North America, and as such the study represents an important contribution both methodologically and substantively. Applying the sigma dissimilarity measure developed by the authors to find the geographically nearest analog is a novel method with intriguing promise for future research. Further, identifying and mapping analogs for a large number of cities constitutes an important educational and informational device – the online mapping tool is particularly effective.

Nonetheless, at present the authors overstate the ability of this paper and the climate-analog method to inform adaptation strategies for the urban areas they studied. By being more precise and measured about the contribution of the study, the authors could improve this paper into a greater contribution.

General comments:

- My main concern is that the authors conflate the term “vulnerability” with what some scholars call “exposure” (pg. 2). The authors claim that they are identifying analogs for the future “vulnerability” of each city in the dataset. But vulnerability means more than simply the climatic conditions of a place, as it has social and political dimensions. Climate change vulnerability is commonly understood to include a) exposure, or the climatic conditions of a place or system; b) sensitivity, or the degree to which the place or system will be affected by climatic conditions; and c) adaptive capacity, or the ability of a place or system to adjust to the climatic conditions. As I understand it, the present study identifies the future climatic exposure of a place using climate models. The authors should work to clarify the difference between exposure and vulnerability.

- o Definition of vulnerability taken from: Adger, W. Neil. "Vulnerability." *Global environmental change* 16.3 (2006): 268-281.

- The consequence of conflating vulnerability and exposure is that the manuscript, as it currently stands, overstates this method’s ability to inform policy that protects cities from climate change impacts. It is unclear to the reader how the knowledge of where a climate-analog is located can help inform adaptation strategies. The main problem here is that the political and social conditions between the original city and its climate-analog, for the vast majority of cases, are not comparable. For example, the reality of climate governance in New York is incomparable to that of Jonesboro, Arkansas. This is an extreme example, but it illustrates how as a reader it is difficult to see how knowledge of climate-analogs represents a policy-learning opportunity.

- In response, the authors should work to clarify how they propose that cities could learn from their climate-analogs.
- The authors should also focus on highlighting the methodological contribution of the study and the importance of the online mapping tool as a novel educational and informational tool, rather than encouraging decision-makers to learn about adaptation strategies from their climatic analogs. To do this, the authors could conduct a more complete overview on literature of cognition and perception of climate risks. They could also explain how their visualization tool could make temporally and geographically distant climate change impacts more concrete for the public.
- My last comment is not necessary to improve the paper for publication, but would improve the effectiveness of the online mapping tool: the authors could identify a place name located within the average climate-analog area (for both scenarios, business-as-usual and emissions reduction). This would facilitate comparison between place names rather than between a place name and a general area on the map, e.g. "Montreal's weather in 2080 will be somewhat similar to Philadelphia's if we don't reduce emissions".

Specific questions and/or comments:

- The title could be more precise and use the actual date of the projections, 2080, instead of "in the late 21st century".
- Pg. 3: the authors identify "natural daily and annual variability of local weather and climate experienced by people" as the greatest barrier to public recognition of climate change. The literature on the cognition and perception of impacts is well-developed beyond this argument and, recognizing space limitations, should be dealt with in a more nuanced way in this paper, or this sentence should be changed. The following papers might provide a starting point:
 - o Weber, Elke U. and Paul C. Stern. 2011. "Public Understanding of Climate Change in the United States." *American Psychologist* 66(4):315.
 - o van der Linden, Sander et al. 2017. "Culture versus Cognition Is a False Dilemma." *Nature Climate Change* 7(7):457.
 - o van der Linden, Sander, Edward Maibach, and Anthony Leiserowitz. 2015. "Improving Public Engagement with Climate Change: Five 'best Practice' Insights from Psychological Science." *Perspectives on Psychological Science* 10(6):758–63.
- Pg. 15: Why were urban areas selected on geographic size rather than population size? I assume this has to do with the sampling of future climate values? The authors should make their explanation explicit.
- Pg. 15: What does a "broad geographic sample" of cities mean to the authors, what criteria did they use to operationalize this, and how were these criteria justified?

- Pg. 15: As I understand it, this is the first study to apply Mahony et al.'s (2017) sigma dissimilarity method for geographically mapping climate analogs. Is this correct? If so, the authors should make this distinction explicit to highlight their contribution.

Reviewer #1 (Remarks to the Author):

This is a very welcome and interesting contribution that employs a state-of-the-art climate matching approach to explore future climate analogs of 540 urban areas in North America, under two emissions scenarios and 27 climate models. The paper is innovative and clearly push forward the field of climate analogs research. Such research is also particularly attractive and relevant for a lay audience, and has a very useful online tool that can be employed by stakeholders and the general public to explore future climate analogs of all the 540 cities. In short, I think this paper fits very well the high-quality standards and the scope of Nature Communications. Yet, I do have a number of issues with the current version, that I think need to be addressed.

Thank you for your careful review and thoughtful comments. In addressing your concerns, we feel our manuscript is vastly improved.

Most of them relate to the fact that you oversell the potential applications of your approach as well as its results. For instance, you state in the abstract (L35) that “our results suggest that through the interactions with infrastructure, resources, and human health, climate change may have profound implications for urban populations”. This, in my opinion, does not hold true. This research identifies the climate analogs of 540 urban areas, based on temperature and precipitation only. Infrastructure, resources, health, etc. are not touched upon in the paper, hence such concluding (and very broad) statement should not be made. I suggest to stick to the main goal of this paper, which is (as you stated L33): “providing an intuitive means of raising public awareness and informing adaptation strategies”.

Thank you for this useful insight. We have tempered our language throughout the manuscript to focus on assessing and communicating climate change implications using climate analogs.

Hereunder are specific comments and questions that I would like you to address. These are mostly minor comments.

L19: why “vulnerability assessments”? What is the link with climate analogs? Are climate analogs really useful for vulnerability assessments?

We agree that “vulnerability” is a poor word choice and that climate analogs do not actually quantify vulnerability (Adger 2006). Throughout the manuscript, we now avoid the use of this term, opting for “exposure” instead as necessary.

L20: climate analogs have also been used to match the current climate of a location with the past climate of another location, e.g. see Beniston, 2013 (<https://doi.org/10.1002/joc.3804>). This might be worth mentioning.

Thank you for making us aware of this paper. Given word limitations and the need to focus the abstract on the primary topic of our study (future climate), we feel that mentioning past climate here may be distracting. However, we now cite Beniston (2014) on L57 where we state:

“Climate-analog mapping is a statistical technique that quantifies the similarity of a location’s climate relative to the climate of another place and/or time”

L26: is RCP8.5 a business-as-usual scenario? It is often rather described as a high-end scenario. Also, for the readers that are not familiar with the RCPs, it might be good to avoid referring to the RCP in the abstract (such a paper might be of interest to a lay audience).

Great point. We have removed mention of RCPs and edited L27-28 as follows:

“We find that if emissions continue to rise throughout the 21st century, climate of North American urban areas...”

L30: in the same sentence you say “under business-as-usual emissions” and “varying by future climate scenario”, that is confusing. Please clarify.

We agree this is confusing. We removed mention of climate scenarios here and moved it to a line above so that the reader knows we are referring to multiple scenarios. The relevant text now reads (starting on L24):

“For 540 urban areas in North America and 54 future climate scenarios, we used climate-analog mapping to identify the geographic location that has a contemporary climate most similar to each urban area’s expected climate by the 2080’s. We find that if emissions continue to rise throughout the 21st century, climate of North American urban areas will shift considerably to become, on average, most like the contemporary climate of locations 850 km away and mainly to the south, with the distance, direction, and degree of similarity to the best analog varying by region and assumptions regarding future climate.”

L80: These are novel future climatic conditions for the region investigated only (North-America), but these may well exist elsewhere.

This is true - better matches (i.e., lower novelty) might be found if we expanded the study domain (but see lines 168-171 for why this is not always an ideal solution in the context of the utility of analogs for informing the public). We edited this line to read (Now begins on L83):

“Values greater than 4σ represent extreme differences between future climate and contemporary climate *within the study domain* (Fig. S1), which we interpret as novel future climatic conditions and a poor analog.”

Note that we also have included a new supplementary figure (Fig. S1) showing the extent of the study domain, which we reference on L84.

L88: This is a very well designed and useful online tool. In my opinion, it should be better advertised than through a simple link. I wonder to which extent this online tool can be embedded within the final manuscript online.

Thanks for the words of encouragement. We struggled somewhat with how best to highlight the online tool, which we feel is crucial for conveying our findings to the public. We now mention the web application in the abstract (L34-37) and have added a paragraph of discussion (L214-228) that describes the tool. Hopefully these edits will help better advertise the online tool, and if accepted, we hope to work closely with the editorial office to better interface the online application with the publication.

L120: Please explain shortly why the greatest geographic distances are expected for Eastern US cities. Is it because their future climate show larger changes, or because of their geographical location, or...?

We have added some text to explain why we think the greatest distances occur mainly in the east. In their analysis, Mahony et al. (2016) found that lower topographic positions were associated with higher novelty. We suspect this effect is at work in our study & likely translates into greater distances to the best analog. We have edited the text (L132-135) as follows:

“The greater distances to the best analog for eastern urban areas likely reflect the influence topographic position on climate. In short, in regions of high relief, such as portions of western North America, lower elevations can provide analogs to higher elevation climates that are expected to become warmer and drier.”

L122: Fig2: is there a specific reason explaining climate analogs of some North-Western US cities are located on the North, unlike all the other cities' analogs? Would be nice to have this explained in the paper.

As for the greater geographic distances in the east, we feel this pattern again reflects the influence of topography on climate. We edited the text on L135-140 to read:

“The average cardinal direction to the best analog was south-southwest and did not differ appreciably between the RCP4.5 and RCP8.5 emission scenarios (200.4° vs. 201.7° respectively). However, for some cities in the Northwest, and especially for RCP4.5, the closest analog was to the north (Fig. 2a), also likely reflecting the influence of topography on the location of the best climatic analog.”

L135: Again, it would prove useful to have some more detailed explanations here. Why such geographical patterns? Why cities west of the Rocky Mountains have “better” climate analogs than cities of Western US and Florida? Is it due to different warming rates, or simply to the geographical border of the computational grid?

We have added text on L154-158 that now reads:

“In contrast, for most urban areas along the western and southeastern coasts, there are no representative contemporary climatic analogs anywhere in the study domain, which likely reflects a combination of the lower topographic position of urban areas in these regions, the nature of forecasted climate change, and the pool of contemporary climates available within the study domain (Fig. S1) to serve as analogs.”

L144: Although the coordinates of the study domain are given in the Methods section, I find it difficult to picture it. It might be useful to have a map showing the study domain. If Fig.2 shows the study domain, then it could be stated in the legend.

Thanks for this suggestion. We have added a new supplementary figure (Figure S1) that shows the extent of the study domain relative to the urban areas included in the analyses.

L181: You quantified and communicated the potential magnitude of changes in climatic conditions, but not the magnitude of “future vulnerabilities” of the cities. These are two different things.

Thanks for pointing this out. We edited L205-209 to read as follows:

“We quantify and communicate the potential exposure of North American urban populations to changes in climatic conditions by identifying where on the landscape similar contemporary climates reside, but we ignore additional stressors, notably supplemental warming associated with urban heat island effects, sea-level rise, and extreme events.”

L186: In the same vein, one can doubt that such an approach facilitates the understanding of the nature of future climate impacts. Can you provide a bit more explanation here? An interesting reference to add is Jylhä et al. 2010 (<https://doi.org/10.1175/2010WCAS1010.1>), in which they actually tested (with a questionnaire) whether climate analogs are a useful tool for communicating future climate impacts.

We edited much of the text in this section to bring more attention to the literature on the use and utility of climate analog analysis. Based on our literature search, we could not find many studies (other than Jylhä et al. 2010) that have formally

tested how well climate-analog mapping works as a tool to raise awareness or educate. We now cite this paper in a new paragraph (L214-228) that focuses on the potential utility of the method and which also advertises the online tool:

“Meaningful visualization and communication are considered key components in raising public awareness of climate change, and approaches similar to those employed in this study have been found to be effective for conveying climate change information. Successful policy implementation on climate change requires public support, which is predicated on fluent communication between scientists, the public, and decision makers. Therefore, in supplement to the summaries presented here, we developed an online, interactive ‘Urban area climate analog mapper’ application (<https://fitzlab.shinyapps.io/cityapp/>) that allows the general public, educators, decisions makes, and stakeholders to explore in greater detail results for all urban areas and climate scenarios. The web application maps the locations of the best contemporary climatic analogs and their sigma dissimilarity, as well as climate similarity surfaces (as in Fig. 1) showing spatial variation in the strength of analogy between future and contemporary climates for each urban area. More research is needed to formally assess the efficacy of these visualizations for communicating climate change information and changing perceptions across a variety of potential end users.”

L190: This discussion on the usefulness (and the limitations) of the climate analogs approach for adaptation/urban planning is missing two important papers: Kellett et al. 2015 (<https://doi.org/10.1016/j.landusepol.2014.11.022>), which points out the limitation of such an approach; and Hallegatte et al. 2007 (<https://doi.org/10.1007/s10584-006-9161-z>), in which authors employ the climate analog method to assess future economic impacts in urban areas.

Thank you for making us aware of these studies, which we now cite in a heavily edited paragraph beginning on L229:

“Studies on the utility of analogs for informing policy (Kellett et al. 2015) and assessing economic impacts (Hallegatte et al. 2007) have been mixed. In addition to climate similarity, economic differences and social and political norms related to pro-environmental behavior and decision making are likely to influence the extent to which climatic analogs can advance policy, adaptation and planning.”

L197: studi study

Fixed.

L198-203: I think that this statement is a bit overselling the paper (and there is no need, as it is already very innovative). The climatic variables that you employed are not determinants for weeds, insect pests, diseases, etc, which also have a range of climatic conditions to which they can spread (and this is not reflected in the climate analog approach). I agree that applying the

climate analogs for these issues is worth exploring (it has already been applied for forestry or wine for instance), but your paper does not do this. I would rather rephrase to reflect the potential applications of the climate analog approach, but with different climatic variables (i.e. oriented towards the goal of the study, e.g. specific to a given type of agriculture, to a given disease, etc.). In the same vein, I would suggest to focus the discussion on the benefits of your approach for the design of urban adaptation strategies, climate-resilient infrastructure, etc (as touched upon in L188-197).

Thank you for pointing this out. We agree we may have oversold our results here, but we disagree that the climate matching method is not useful for determining source regions for new organisms. Climate-matching techniques incorporating the sort of variables we used have been used for decades for this very purpose, including Mahalanobis distance that forms the basis of sigma dissimilarity.

We have edited this paragraph (starting on L244) to focus it more on the method than on our specific results and to point the reader to a relevant example from the literature. All that being said, we are happy to remove this paragraph if needed.

“Beyond assessing exposure and communicating aspects of climate change itself, climate-analog mapping also can inform how climate change could impact agricultural and natural systems upon which humans depend. For example, climatic similarity between regions is often predictive of the success of introduced organisms (Thuiller et al. 2005). In this sense, climate analog mapping could be used to identify potential source regions for new, potentially problematic organisms, including weeds, insect pests, and diseases that could impact human health, wildlife, and agriculture. For example, our results suggest northeastern cities by the 2080’s could support species now limited to the southeastern U.S. In instances where these new residents are disease vectors (e.g., Asian Tiger mosquitos and West Nile Virus), urban areas could experience increased incidence of disease. However, as for other applications of climate analogs, climate novelty challenges our ability to use current patterns to anticipate potential ecological responses and threats.”

Thuiller et al. (2005) Niche-based modelling as a tool for predicting the risk of alien plant invasions at a global scale. *Glob. Chang. Biol.* 11, 2234–2250.

L233: Needs more clarification on this rather non-transparent weighting of the contribution of the 12 individual climate variables. Is there one variable that weighs much more than the other? How much dissimilarity for a given variable is allowed? For instance, if the min/max seasonal temperatures fit perfectly with the analog’s temperature regime but the precipitation regimes show a very poor match, is the sigma dissimilarity still low? This needs some transparent clarification.

Sorry for any confusion here. Mahalanobis distance gives equal weight to all variables (notwithstanding the scaling to interannual climatic variation - see Mahony et al. 2017 for details), but corrects for the inflation of distance that occurs when variables are not orthogonal (which is not needed when calculating Euclidean distances using x,y,z coordinates given that each axis is on the same scale and are orthogonal). In short, a naive distance would count two perfectly correlated variables twice. We edit the text to make this more clear (starting on L278):

“Sigma dissimilarity is derived from the Mahalanobis distance, which is a multivariate distance that is independent of the scales of the climate variables and which weights the contribution of individual climate variables to the statistical measure of distance according to their collinearity with all other variables (i.e., Mahalanobis distance corrects for the inflation of distance that occurs when variables are not orthogonal).”

L257: are these projections bias-corrected? If not, this should be at least acknowledged (as well as the uncertainties associated with non-bias corrected climatic projections).

The short answer is yes, they are bias-corrected. We have added text (starting on L305) to make this clear and a citation describing the methods used.

Longer answer: The climate projections are created using a “delta method” or “change-factor method”, whereby (1) climate anomalies are calculated between present day *simulations* and future simulations (which removes and systematic bias in the model); (2) the resulting anomalies are resampled to the desired resolution of the observational data set using spatial interpolation techniques; and (3) the anomalies are then added to baseline current climate observations. This method has been widely employed in climate change research given that it is simple and fast and therefore the most feasible for downscaling of a large number of climate simulations.

L280: what does “2080s” stand for? Is it 2070-2090? 2060-2100? Please clarify

The source of the future climate data used in our study defined 2080s as the mean of the period 2070-2099. We now cite a white paper describing the methods used and have added edits in several places throughout the manuscript to make this more clear. For example:

“Projections of future climate for the 2080’s (30-year running mean of the period 2070-2099) at 30 arc-second resolution were obtained from the...”

Citation added: Ramirez-Villegas, J. & Jarvis, A. Downscaling global circulation model outputs: the delta method decision and policy analysis Working Paper No. 1. (2010).

L281: this is a critical point of the method. I initially thought that you computed the location of the climate analog for each climate model, and then employed a mapping approach to define the location that fits best the 27 climate analog of a given city (e.g. the location that minimizes the distance between each of the 27 analogs' location). It seems however that you averaged the climate data over the 27 ESM, and then defined afterwards the climate analog. This is still a valid approach, but you need to be more critical and to acknowledge that there is other ways of computing analogs for an ensemble of climate models.

Thanks for pointing this out. We have edited this section to inform the reader that our averaging our approach is not the only way the averaging could have been done:

“We also averaged temperature and precipitation forecasts from each of the 27 scenarios across each RCP to create an ensemble mean projection for that RCP (as opposed to first calculating sigma dissimilarity for each forecast and averaging second).”

Reviewer #2 (Remarks to the Author):

This article demonstrated present-day climatic analogs for 540 North American urban areas in 2080. Climate analogue has been used for more than a century and applied to future cities under a changing climate for several decades. The number of the target cities in this article are larger than previous articles but no scientific novelty is seen except for this large number. The target of the audiences is vague since there are many decision makers. Therefore, this article in the present form is not recommended for publication in this journal. There are also some scientifically unclear points mentioned below. By responding these points, this article will improve in scientific contents.

Thank you for your helpful comments. As highlighted by the other two reviewers, the novelty of the work is also in the use of a more robust method for estimating the similarity between climates and the large number of climate scenarios considered. Lastly, we also developed and report an online web application accessible by anyone with web access.

I do not understand why the distance and the cardinal direction is important in this context: “RCP8.5 (849.8 km) as compared to RCP4.5 (514.4 km)” and “the cardinal direction did not differ appreciably between the RCP4.5 and RCP8.5 emission scenarios (200.4° vs. 201.7° respectively)”

We feel that by providing summaries of the distance and cardinal direction (as averages across models within each RCP), we can make several important points. First, by reporting average distance, we provide a sense of the magnitude of exposure. Second, by reporting cardinal direction, we show that the results confirm intuition - most analogs are to the south in a warming climate. And third, we summarize how the results are different (distances) and similar (directions) between the two RCPs.

Methods

Both the Worldclim and NOAA station data are used in this study. The climatological mean values of the NOAA station data are not always the same. You should show the difference between the two climatologies.

We apologize for any confusion here, but we are not sure we fully follow this comment. We use the Worldclim dataset to describe mean climate for the contemporary period (1960-1990). We use the NOAA data to describe interannual climatic variability for each urban area across the same period. The NOAA data are used to appropriately scale the climate distances.

How do you prepare the future climate values? Please describe it since the model has biases in all variables.

We now cite a paper describing the preparation of the future climate variables: Ramirez-Villegas, J. & Jarvis, A. Downscaling global circulation model outputs: the delta method decision and policy analysis Working Paper No. 1. (2010).

We also have added more details on L305-316.

In brief, and as detailed in a response to a comment by R1 above, these future climate surfaces are downscaled and debiased using a “delta method” or “change-factor method”, whereby (1) climate anomalies are calculated between present day *simulations* and future simulations; (2) the resulting anomalies are resampled to the desired resolution of the observational data set using spatial interpolation techniques; and (3) the anomalies are then added to baseline current climate observations. This method has been widely employed in climate change research given that it is simple and fast and therefore the most feasible for downscaling of a large number of climate simulations.

In search of present-day climate analogue, three variables: two air temperatures and precipitation. The climate analogue may be weakly influenced by precipitation. The two air temperatures are projected to rise by almost same degrees and do not provide independent information. It is better to use mean air temperature or to use weighting during the search of climate analogues.

Sorry for any confusion. We actually used 12 climate variable, which we make more clear on L88-89:

“We calculated sigma dissimilarity using minimum and maximum temperature and total precipitation for the four climatological seasons (12 climate variables total).”

The Mahalanobis distance method we employed accounts for correlations (non-independence) between variables and treats all variables equally. However, note that variables are scaled as needed by their interannual climatic variation such that variables that have high historic variation contribute less than variables that have been more consistent.

These three combinations may be justified for a specific purpose such as crop productions, please provide the justification for the selection.

We thought a lot about variable selection and tried different combinations and subsets of variables. Our goal was to select a minimum set of variables that adequately described climate and which tended to align with how humans perceive climate. Some variables of interest (e.g., humidity, cloud cover, etc) were not sufficiently available to support all aspects of our analyses. For these reasons, we selected a set of 12 variables that quantified variation in temperature and precipitation across the four seasons.

Why is the year of 2080 is selected? The climate analogue analysis should use the same temporal average period. The method in this study use present-day climatological values and single-year values but with mutli-model ensemble. The variability of multi-model ensemble is not ICV but the variability of multi-model ensemble or uncertainties in future climate projections. Please provide the justification if both are interchangeable.

Sorry for any confusion here. We referred to “year 2080” in the manuscript, when in fact we used the 30-year mean for the 2080’s, defined as 2070-2099. This matches the extent of the temporal averaging period of 30 years for the contemporary climate normals (1960-1990). We have edited the text to make this more clear & now consistently use the term “the 2080’s” instead of “year 2080”.

In Fig. 4, you identify the climate analog for each model’s projection. For this case, you compared observed climatology with projected single year value. Please provide the justification.

See previous comment.

I believed that use of climatological values of each model provide more information about uncertainties of climate analogues in future climate projections.

We agree, though showing all scenarios for many cities is not practical in print. For this reason, and to provide public access, we created the online tool where individuals can explore results for all cities and scenarios in detail.

Figures S1 and S2 shows large discrepancies in values in some cities between present-day climatological values and future projected values, especially in precipitation. These discrepancies are acceptable?

Figures S1 and S2 show two types of differences. First, the boxplots show how much temperature and precipitation are predicted to change in 4 eastern and 4 western cities. Second, they show how these contemporary and future values compare to the climate of the best analog. Differences between present-day and future climate reflect projected changes in climate - in short they are arise entirely from differences in contemporary climate and the climate forecasts. Differences between future climate and the best contemporary climate analog reflect *climate novelty*, not discrepancies resulting from the statistical methodology.

Reviewer #3 (Remarks to the Author):

Article summary and contribution:

The study identifies and maps climate-analogs for 540 urban areas in the United States and Canada. The authors find that under business-as-usual emissions scenarios, the climate of North American urban areas will shift, on average, 928km away and mainly to the south. The authors also link to a novel online interactive mapping tool to visualize the climate-analogs for each city.

As far as I know, this is the first large-n analysis of climate-analogs in North America, and as such the study represents an important contribution both methodologically and substantively. Applying the sigma dissimilarity measure developed by the authors to find the geographically nearest analog is a novel method with intriguing promise for future research. Further, identifying and mapping analogs for a large number of cities constitutes an important educational and informational device – the online mapping tool is particularly effective.

Thank you for your useful thoughtful comments. We have attempted to address all your concerns and feel our manuscript is much improved as a result.

Nonetheless, at present the authors overstate the ability of this paper and the climate-analog method to inform adaptation strategies for the urban areas they studied. By being more precise and measured about the contribution of the study, the authors could improve this paper into a greater contribution.

We agree and note that R1 made a broadly similar assessment. We have tempered our language throughout the manuscript to focus on assessing and communicating climate change implications using climate analogs.

General comments:

- My main concern is that the authors conflate the term “vulnerability” with what some scholars call “exposure” (pg. 2). The authors claim that they are identifying analogs for the future “vulnerability” of each city in the dataset. But vulnerability means more than simply the climatic conditions of a place, as it has social and political dimensions. Climate change vulnerability is commonly understood to include a) exposure, or the climatic conditions of a place or system; b) sensitivity, or the degree to which the place or system will be affected by climatic conditions; and c) adaptive capacity, or the ability of a place or system to adjust to the climatic conditions. As I understand it, the present study identifies the future climatic exposure of a place using climate models. The authors should work to clarify the difference between exposure and vulnerability. Definition of vulnerability taken from: Adger, W. Neil. "Vulnerability." *Global environmental change* 16.3 (2006): 268-281.

We read the Adger (2006) paper (thank you for this reference) & fully agree that this was poor word choice on our part. We have removed the use of “vulnerability” throughout the manuscript and have replaced with “exposure” as appropriate.

The consequence of conflating vulnerability and exposure is that the manuscript, as it currently stands, overstates this method’s ability to inform policy that protects cities from climate change impacts. It is unclear to the reader how the knowledge of where a climate-analog is located can help inform adaptation strategies. The main problem here is that the political and social conditions between the original city and its climate-analog, for the vast majority of cases, are not comparable. For example, the reality of climate governance in New York is incomparable to that of Jonesboro, Arkansas. This is an extreme example, but it illustrates how as a reader it is difficult to see how knowledge of climate-analogs represents a policy-learning opportunity.

In response, the authors should work to clarify how they propose that cities could learn from their climate-analogs.

We have tempered our language regarding this point and cite a few papers that have explored the use of climate analogs for informing adaptation / policy. We now focus the paper on the use of climate analogs for communicating climate change rather than on their use in informing policy / adaptation.

The authors should also focus on highlighting the methodological contribution of the study and the importance of the online mapping tool as a novel educational and informational tool, rather than encouraging decision-makers to learn about adaptation strategies from their climatic analogs. To do this, the authors could conduct a more complete overview on literature of

cognition and perception of climate risks. They could also explain how their visualization tool could make temporally and geographically distant climate change impacts more concrete for the public.

Thank you for pointing this out. We have read several relevant studies on perception of climate risks, which we summarize and/or cite as appropriate. We also have refocused portions of the Introduction and the Discussion to discuss how climate analog mapping might help address barriers to public awareness of climate change, citing relevant literature as needed. Lastly, we have added text to better advertise the contribution of the online mapping tool.

My last comment is not necessary to improve the paper for publication, but would improve the effectiveness of the online mapping tool: the authors could identify a place name located within the average climate-analog area (for both scenarios, business-as-usual and emissions reduction). This would facilitate comparison between place names rather than between a place name and a general area on the map, e.g. "Montreal's weather in 2080 will be somewhat similar to Philadelphia's if we don't reduce emissions".

We have toyed extensively with this idea and actually implemented it in a previous version of the web application. There are pros and cons - most notably that sometimes analogs are not near a city with name recognition. We agree that this idea warrants further exploration.

Specific questions and/or comments:

The title could be more precise and use the actual date of the projections, 2080, instead of "in the late 21st century".

Sorry for any confusion here. For simplicity's sake, we used "2080" when discussing the future. However, the future climate scenarios actually represent a 30-year running mean of the period 2070-2099. For this reason, we feel using "late 21st century" in the title is more appropriate. We have added edits in several places throughout the manuscript to make this more clear (i.e., changing "year 2080" to "the 2080's and noting the 30-year averaging period).

Pg. 3: the authors identify "natural daily and annual variability of local weather and climate experienced by people" as the greatest barrier to public recognition of climate change. The literature on the cognition and perception of impacts is well-developed beyond this argument and, recognizing space limitations, should be dealt with in a more nuanced way in this paper, or this sentence should be changed. The following papers might provide a starting point:

Weber, Elke U. and Paul C. Stern. 2011. "Public Understanding of Climate Change in the United States." *American Psychologist* 66(4):315.

van der Linden, Sander et al. 2017. "Culture versus Cognition Is a False Dilemma." *Nature Climate Change* 7(7):457.

van der Linden, Sander, Edward Maibach, and Anthony Leiserowitz. 2015. "Improving Public Engagement with Climate Change: Five 'best Practice' Insights from Psychological Science." *Perspectives on Psychological Science* 10(6):758–63.

Thank you for making us aware of this literature. We have read these papers as well as several others. We have edited portions of the Introduction and Discussion to frame the study more in the context of visualizing and communicating climate change risks and now cite this literature as appropriate.

Pg. 15: Why were urban areas selected on geographic size rather than population size? I assume this has to do with the sampling of future climate values? The authors should make their explanation explicit.

Please see response to the next comment.

Pg. 15: What does a "broad geographic sample" of cities mean to the authors, what criteria did they use to operationalize this, and how were these criteria justified?

We edited this section for clarity and to correct the description of the methods used to select urban areas. The original text was copied from a previous draft and reflected a different approach for selecting urban areas than that ultimately used for the results reported in this study. In short, we selected purely based on data requirements related to: (1) size of the city, (2) climate records, and (3) spatial heterogeneity in climate across the urban area, not on geographic representativeness. The updated methods now read:

"To ensure a large enough sample size for statistical analyses (where n is the number of $\sim 1 \times 1$ km² pixels within the urban area), we selected urban areas with a geographic area greater than ~ 50 km². Urban areas were removed from further consideration if they lacked sufficient (i) historical weather records or (ii) spatial heterogeneity in climate as necessary for model fitting. We also lumped multiple adjacent urban areas, for example to combine suburbs with an adjacent city (e.g., Anchorage, AK and Northeast Anchorage, AK), resulting in 540 urban areas total."

Pg. 15: As I understand it, this is the first study to apply Mahony et al.'s (2017) sigma dissimilarity method for geographically mapping climate analogs. Is this correct? If so, the authors should make this distinction explicit to highlight their contribution.

This is the first study to apply Mahony et al.'s method to the idea of locating and mapping climate analogs for urban areas. In Mahony et al., the authors mapped climate novelty for each pixel in North America, with a focus on describing sigma

dissimilarity, a more robust statistical method for estimating climate novelty and its drivers.

We were nearing submission of an earlier version of this study based on Mahalanobis distance (without the additional step of calculating sigma dissimilarity) when the Mahony et al (2017) paper appeared in *Global Change Biology*. Recognizing the utility of translating Mahalanobis distance into sigma,, we redid all of our analyses (several months of computation time) using the sigma dissimilarity approach.

REVIEWERS' COMMENTS:

Reviewer #1 (Remarks to the Author):

First of all, let me thank you for the elaborate response to my comments on the previous version. You provide very clear arguments and good solutions to all the concerns I had. As a result, I have no further substantive comments on the substance of the manuscript and think that the manuscript is almost ready for publication. I am looking forward to seeing the published version of this interesting and very useful contribution to the advancement of the use of climate analog mapping for awareness raising. I also hope that the online tool that accompanies this paper will be widely advertised.

I have only a few minor comments (which are mostly suggestions only) that you may want to consider for the final version of the manuscript:

L27-31: This sentence is rather long; can you cut it into two?

L33: “novel climate” instead of “novel future climate”?

L43: Ref n°5: is there a similar study that has been conducted over Northern American cities only (with a larger sampling)? If so, that would be relevant to cite it here instead of Ref. n°5

L53: replace “rewards” by “opportunities”?

L68: “have been recently developed”

L76: Using 1960-1990 (i.e. 1970s) as representative of today’s climate is a quite important limitation that may lead to over-estimating the climate shift. It may be worthy to mention this in the method section.

L90: Spell out RCP (if this is a requirement of the journal)?

L91: “54 future couples of climate model-scenario” – otherwise the reader can get confused between “emission scenario” and “climate scenario”.

L139: for these Northern analogs, did you look at the difference in elevation? If yes, and if it plays an important role, it should be mentioned here (which would specify “topography”).

L251: It should be added here that the climatic variables you employed might not cover the whole range of bio-climatic variables that determine whether species can live and propagate at a given location.

L262: Could you elaborate a bit more on how this could be done? A number of climate analog studies have concluded with the same recommendation, but none has actually detailed how this

could be done. I think it would be very beneficial to shortly explain here (two or three sentences) how such assessment of the utility for awareness raising of climate analog mapping can be done.

L321: same issue as in the main text with the term “climate scenario”

Reviewer #2 (This reviewer only left remarks to the editor)

Reviewer #3 (Remarks to the Author):

In my judgement, the authors have addressed my comments appropriately and clarified where I requested them to.

Reviewer #1 (Remarks to the Author):

First of all, let me thank you for the elaborate response to my comments on the previous version. You provide very clear arguments and good solutions to all the concerns I had. As a result, I have no further substantive comments on the substance of the manuscript and think that the manuscript is almost ready for publication. I am looking forward to seeing the published version of this interesting and very useful contribution to the advancement of the use of climate analog mapping for awareness raising. I also hope that the online tool that accompanies this paper will be widely advertised.

Thank you again for your careful review and thoughtful comments.

I have only a few minor comments (which are mostly suggestions only) that you may want to consider for the final version of the manuscript:

L27-31: This sentence is rather long; can you cut it into two?

Edit made.

L33: “novel climate” instead of “novel future climate”?

Edit made.

L43: Ref n°5: is there a similar study that has been conducted over Northern American cities only (with a larger sampling)? If so, that would be relevant to cite it here instead of Ref. n°5

We opted to keep the cited reference as we are speaking in general terms here.

L53: replace “rewards” by “opportunities”?

Edit made.

L68: “have been recently developed”

Fixed.

L76: Using 1960-1990 (i.e. 1970s) as representative of today’s climate is a quite

important limitation that may lead to over-estimating the climate shift. It may be worthy to mention this in the method section.

We agree that the results could be sensitive to the time period used for contemporary climate. We added some text to the methods to make this point.

L90: Spell out RCP (if this is a requirement of the journal)?

Edit made.

L91: “54 future couples of climate model-scenario” – otherwise the reader can get confused between “emission scenario” and “climate scenario”.

Edited to clarify.

L139: for these Northern analogs, did you look at the difference in elevation? If yes, and if it plays an important role, it should be mentioned here (which would specify “topography”).

We did not specifically look at the role of elevation. However, Mahony et al. (2017) did in their study, which we reference in this section.

L251: It should be added here that the climatic variables you employed might not cover the whole range of bio-climatic variables that determine whether species can live and propagate at a given location.

Good point. Edit made to read: “In this sense, climate analog mapping based on an ecologically relevant set of variables could be used to identify potential source regions ...”

L262: Could you elaborate a bit more on how this could be done? A number of climate analog studies have concluded with the same recommendation, but none has actually detailed how this could be done. I think it would be very beneficial to shortly explain here (two or three sentences) how such assessment of the utility for awareness raising of climate analog mapping can be done.

It is difficult to get into the specifics here given that there are many methods that could be used to assess the utility of climatic analog mapping for raising awareness. However, we feel that in general, the most fruitful way forward would be to work with experts in the fields of climate change education, psychologists, and social scientists to design projects to rigorously test climatic analog mapping as an tool. We have edit the final line of the manuscript to make this more clear.

L321: same issue as in the main text with the term “climate scenario”

We state parenthetically that the 54 climate scenarios arise from the combination of 27 earth system models crossed with two RCPs, i.e., 27 earth system models x 2 RCPs.